# Polyhydroxyalkanoate Nanoparticles for Pulmonary Drug Delivery: Interaction with Lung Surfactant

**DOI:** 10.3390/nano11061482

**Published:** 2021-06-03

**Authors:** Olga Cañadas, Andrea García-García, M. Auxiliadora Prieto, Jesús Pérez-Gil

**Affiliations:** 1Department of Biochemistry and Molecular Biology, Faculty of Biology, Complutense University, 28040 Madrid, Spain; ocanadas@quim.ucm.es (O.C.); Andrea19@ucm.es (A.G.-G.); 2Research Institute “Hospital Doce de Octubre (imas12)”, 28041 Madrid, Spain; 3Department of Microbial and Plant Biotechnology, Centro de Investigaciones Biológicas Margarita Salas (CIB-CSIC), 28040 Madrid, Spain; auxi@cib.csic.es

**Keywords:** DPPC, POPG, PHA, dynamic light scattering, lipid monolayers, relaxation kinetics, epifluorescence microscopy, polyhydroxyalkanoates, PHA nanoparticles

## Abstract

Polyhydroxyalkanoates (PHA) are polyesters produced intracellularly by many bacterial species as energy storage materials, which are used in biomedical applications, including drug delivery systems, due to their biocompatibility and biodegradability. In this study, we evaluated the potential application of this nanomaterial as a basis of inhaled drug delivery systems. To that end, we assessed the possible interaction between PHA nanoparticles (NPs) and pulmonary surfactant using dynamic light scattering, Langmuir balances, and epifluorescence microscopy. Our results demonstrate that NPs deposited onto preformed monolayers of DPPC or DPPC/POPG bind these surfactant lipids. This interaction facilitated the translocation of the nanomaterial towards the aqueous subphase, with the subsequent loss of lipid from the interface. NPs that remained at the interface associated with liquid expanded (LE)/tilted condensed (TC) phase boundaries, decreasing the size of condensed domains and promoting the intermixing of TC and LE phases at submicroscopic scale. This provided the stability necessary for attaining high surface pressures upon compression, countering the destabilization induced by lipid loss. These effects were observed only for high NP loads, suggesting a limit for the use of these NPs in pulmonary drug delivery.

## 1. Introduction

The therapeutic efficacy of inhaled nanoparticles depends on their ability to cross the extracellular and cellular defensive barriers imposed by the lungs. In this sense, the nanoparticles that reach the alveoli first meet the interfacial layers of pulmonary surfactant, a lipoprotein complex that covers the alveolar epithelium and constitutes the first line of defense against inhaled entities. In addition to participating in the innate immune defense of the lung, pulmonary surfactant is crucial to lower surface tension at the air/liquid interface to values close to 0 mN/m, avoiding atelectasis and facilitating respiratory mechanics [1]. The surface activity of pulmonary surfactant is characterized by three essential properties that have to be exhibited during respiration: (i) rapid adsorption at the air/alveolar fluid interface of the material secreted by type II pneumocytes and its transfer to the interface during inhalation, forming a surface-active film; (ii) reorganization of the interfacial film during the interfacial compression that occurs at exhalation, which allows properly packing, further reducing surface tension at the interface to extremely low values and thereby preventing the collapse of alveoli; and (iii) efficient spreading of packed lipids to redistribute laterally during the expansion of the air/liquid interface on subsequent inhalation [2,3,4].

Pulmonary surfactant is composed of 90% lipids, dipalmitoylphosphatidylcholine (DPPC) being the main phospholipid species. This lipid is characterized by saturated acyl chains, which allow it to pack tightly at the air/liquid interface, producing maximum reduction of surface tension at the end of exhalation and stabilization of open lungs [2]. Pulmonary surfactant also contains other phosphatidylcholine species, mainly unsaturated, acidic phospholipids such as phosphatidylglycerol, and cholesterol. In addition to the lipid components, surfactant contains four specific proteins: the hydrophobic proteins SP-B and SP-C that facilitate respiratory mechanics by direct association with surfactant layers and the collectins SP-A and SP-D that mainly participate in the innate immune defense of the lung. The possible interactions between inhaled nanoparticles and surfactant components critically depend on the composition of the nanomaterial, as well as on its size and charge [5,6]. The subsequent adsorption of surfactant lipids and proteins on the nanoparticle surface, forming a lipoprotein corona [7], may inhibit the surface-active and immunomodulatory properties of surfactant by preventing these lipids and proteins from locating at the interfacial film or in the aqueous phase where they perform their functions (reviewed in ref. [6]). On the other hand, the formation of the corona alters the surface properties of the nanoparticles and, therefore, can modify their biodistribution and toxicity [8,9]. For instance, the coating of nanoparticulated gels with a lipoprotein film that contains SP-B favors the vehiculization of encapsulated small interfering RNA [10,11]. Furthermore, recent studies by our group show that surfactant complexes favor the efficient distribution of different drugs along the air-liquid interface, promoting their surface-associated diffusion over long distances [12,13,14].

Among the different biopolymers with biomedical applications, it is worth highlighting the polyhydroxyalkanoates (PHAs). PHAs are a class of polyesters produced by more than 300 Gram-positive and -negative bacteria in sustainable processes characterized by the presence of high carbon concentrations and limiting conditions of other nutrients. This bioplastic accumulates in the form of insoluble spherical inclusions or granules as carbon and energy storage materials. Structurally, PHA polymers are composed of up to 150 different monomers of hydroxylated fatty acids at carbon 3 that are linked through esterification of the carboxyl groups with the hydroxyl groups of the subsequent monomers, progressively elongating the molecule [15,16]. PHAs are characterized by being biodegradable, biocompatible, and very versatile, which is why they have been used in different biomedical applications, both for the development of drug delivery systems and in tissue engineering [17,18]. Furthermore, PHA-based nanoparticles have better bioavailability and encapsulation capabilities as well as less cytotoxicity than other polymeric materials [19,20].

The main objective of this work was to evaluate the potential application of PHA nanoparticles as part of inhaled drug-delivery systems. To that end, we have assessed the possible interaction between PHA nanoparticles (NPs) and pulmonary surfactant lipids. Our results suggest that NPs could be useful for lung drug delivery since the nanomaterial translocates into the subphase without apparently affecting the compression isotherms of DPPC and DPPC/POPG films at lipid/nanomaterial weight ratios below 1:0.33. Larger amounts of NPs slightly fluidized both types of lipid films without altering the ability of DPPC to reduce surface tension to values below 2 mN/m.

## 2. Materials and Methods

PHA, a heteropolymer of poly(hydroxioctanoate-co-hexanoate) (Figure 1), was provided by Bioplastech, Ltd. (Dublin, Ireland). Dipalmitoylphosphatidylcholine (DPPC), palmitoyloleoylphosphatidylglycerol (POPG) (Figure 1), and the fluorescent lipid 1-palmitoyl-2-{6-[(7-nitro-2-1,3-benzoxadiazol-4-yl)amino]hexanoyl}-sn-glycero-3-phosphocholine (NBD-PC) were from Avanti Polar Lipids (Alabaster, AL, USA). The organic solvents, chloroform, and methanol used to dissolve lipids, as well as acetone, which was used to dissolve PHA were HPLC-grade (Thermo Fisher Scientific, Waltham, MA, USA).

### 2.1. Nanoparticle Preparation

For PHA nanoparticle formation, a previously described emulsion-solvent method [21] was used with some modifications. Briefly, PHA solubilized in acetone was added drop by drop to an aqueous solution while the mixture was magnetically stirred at 700 rpm in an Agimatic-N magnetic stirrer (Thermo Fisher Scientific, Waltham, MA, USA) at room temperature to facilitate emulsion formation. The organic solvent was allowed to evaporate at room temperature overnight.

### 2.2. Lung Surfactant Models

To evaluate the interaction of NPs with the different components of pulmonary surfactant, three surfactant models were used: native pulmonary surfactant (NS) purified from porcine lungs obtained from the slaughterhouse, the organic extract (hydrophobic fraction) of pulmonary surfactant (EO), which contains all the surfactant lipids and the hydrophobic proteins SP-B and SP-C, and interfacial films and multilamellar suspensions composed of the main phospholipid species of surfactant, DPPC or a DPPC/POPG 7:3 (*w*/*w*) mixture (PL). Animals used to recover these materials were healthy and subjected to Vet control according to local regulations (Spanish hygiene rules legislation, law articles 83, 85 and 178). Pigs were sacrificed for food and not for the sole purpose of the study.

NS was purified as described [22]. Briefly, porcine lungs were rinsed with 2.5 L of buffer A (5 mM Tris-HCl, pH 7.4, 150 mM NaCl) and cells and debris were removed from the bronchoalveolar lavage fluid by centrifugation at 1000× *g* for 10 min. Next, the supernatant was centrifuged for 1 h at 100,000× *g* and 4 °C using a 70 Ti fixed-angle rotor (Beckman Coulter, Brea, CA, USA). The resulting pellet was homogenized in 16% (*w*/*v*) NaBr 0.9% (*w*/*v*) NaCl and NS was obtained by density gradient centrifugation using a NaBr gradient and homogenization of the resulting surfactant disc in 1.5 mL of 0.9% (*w*/*w*) NaCl.

The hydrophobic fraction of NS was prepared by chloroform/methanol extraction as described by Bligh and Dyer [23]. Briefly, NS was homogenized in a chloroform/methanol/water (1:2:1 by volume) mixture and incubated for 30 min at 37 °C to promote protein flocculation. Next, phase separation was induced by addition of chloroform and water and sample centrifugation at 3000× *g* and 4 °C for 5 min, promoting the separation of the hydrophobic and polar components of NS. Successive lavages with chloroform were performed to maximize the amount of NS components in the hydrophobic phase. The organic phase was collected after each lavage.

To obtain interfacial films and multilamellar suspensions of DPPC and POPG, both lipids were first solubilized in chloroform/methanol 2:1 (*v*/*v*). To prepare multilamellar suspensions, the required amounts of the organic solutions of DPPC and POPG were taken and mixed and evaporated to dryness under a gentle stream of nitrogen, with solvent traces being subsequently removed by evacuation under reduced pressure for 2 h as described in [24]. Then, dry lipid films were hydrated in buffer A at 45 °C.

Total phospholipid concentration in the different samples was estimated by phosphorus quantitation by the method of Rouser [25].

### 2.3. Dynamic Light Scattering (DLS) Measurements

DLS was used to determine the apparent hydrodynamic radius (*R*_H_) and polydispersity (Pd) of PHA nanoparticle suspensions as well as to characterize the effect of surfactant components on the nanomaterial size distribution. To that end, a DynaPro MS/X DLS detector equipped with an 824.7 nm-laser (Wyatt Technology, Santa Bárbara, CA, USA) was used. Briefly, the nanomaterial (1 mg/mL), alone or incubated for 10 min with NS, EO, PL, or DPPC multilamellar suspensions, was diluted 100-fold with milliQ water filtered 10 times with filters of 0.22 μm (Q-Pod, Merck, Darmstadt, Germany) and the hydrodynamic radius of the different components of the sample was calculated by the Stokes–Einstein equation (Equation (1)):(1)$$R_{H}=\frac{k_{B}T}{6\pi \eta D}$$
where *D* is the translational diffusion coefficient, *k*_B_ the Boltzman constant, *T* the temperature, and *η* the viscosity. The same surfactant suspensions analyzed by DLS were exposed to the presence of NPs.

Polydispersity values smaller than 15% were considered to correspond to monodisperse samples.

### 2.4. Monolayer Experiments

Monolayer experiments were performed in a thermostated Langmuir–Blodget trough (total area of 195 cm^2^, 302RB ribbon barrier film balance, NIMA Technologies, Coventry, UK) as previously described [25]. All measurements were performed at 25.0 ± 0.1 °C.

To obtain surface pressure-area isotherms, monolayers of DPPC and DPPC/POPG were formed by spreading 10 µL of the lipid organic solutions (1 mg/mL) onto a buffer A subphase. The solvents were allowed to evaporate for at least 10 min before starting monolayer compression at 50 cm^2^/min. For all the isotherms, an equal number of phospholipid molecules was spread at the interface (13.6 nmol).

In a first approach, and to mimic exposure following inhalation, NPs were deposited onto preformed DPPC and PL films. To that end interfacial films of the corresponding phospholipids were first formed by spreading 10 µL of the organic solutions of the lipids, and only once the organic solvents were evaporated and the interfacial films were equilibrated, different volumes of a PHA nanoparticle suspension at 0.5 mg/mL were deposited by a microsyringe at different places of the Langmuir trough and allowed to interact with the interfacial film for 10 min before the recording of compression isotherms. The amounts of phospholipids and nanoparticles deposited on the air/liquid interface was calculated considering the volumes deposited of each solution and their respective concentrations. After testing the effect of different NP/phospholipid ratios, the experiments in the presence of three different illustrative NP amounts (0.13, 0.33, and 0.67 lipid/nanomaterial weight ratio) are presented in the figures.

Alternatively, in a second approach, we tested the potential effect of the possible direct partitioning of bioplastic polymer molecules into the lipid layers. For this purpose, mixed lipid/PHA monolayers were obtained by deposition of 15 µL of a mixture of 5 µL of the polymer dissolved in acetone, at different concentrations, with 995 µL of the chloroform/methanol lipid solution and the mixture was deposited onto the air/liquid interface. Isotherms of the lipid/polymer mixtures were recorded after solvent evaporation. Control experiments performed to evaluate the potential effect of the presence of traces of acetone at the surface film, by adding equivalent amounts of pure acetone to the lipid solution, indicated that the addition of acetone to the DPPC organic solution did not affect the compression isotherm of DPPC monolayers. Data shown are the mean of six independent measurements.

Compression isotherms were analyzed in terms of the compressibility modulus, $C_{S}^{-1}$, (Equation (2)):(2)$$C_{S}^{-1}=-A{\left(\frac{d\pi }{dA}\right)}_{T}$$
where *A* is the molecular area, *π* the surface pressure, and *T* the temperature.

### 2.5. Relaxation Kinetics

DPPC and PL monolayers were compressed to a surface pressure of 47 or 33 mN/m that was kept constant by automatically adjusting the surface area of the trough through the movement of the ribbon barrier. Once the desired surface pressure was reached, either NPs or buffer was deposited onto the interfacial film. A relaxation curve was obtained by recording the trough surface area during the relaxation period as in [26].

To characterize the effect of NPs on the relaxation kinetics of both lipid films, data were analyzed by fitting to Equation (3):(3)$$-log\left(A/A_{0}\right)=k\cdot \sqrt{t}$$
where *A* and *A*_0_ are the trough areas at a given time *t* and at *t* = 0, respectively, and *k* is the rate of the desorption process [27].

### 2.6. Epifluorescence Microscopy

The effect of NPs on the lateral structure of lipid films was analyzed by epifluorescence microscopy. Briefly, DPPC, alone or mixed with POPG (1 mg/mL final lipid concentration), was incubated with the fluorescent dye NBD-PC (Molecular Probes, Life Technologies, Carlsbad, CA, USA) for 1 h at 37 °C to obtain a final molar ratio dye/surfactant of 1%. Next, 15 µL of the lipid/dye suspension was spread onto the air-liquid interface of the Langmuir–Blodgett trough and the organic solvent was allowed to evaporate for 10 min. Afterwards, the interfacial film was transferred onto a glass coverslip during compression at a constant speed of 25 cm^2^/min using the COVASP LB technique [28]. Traditional LB films prepared at constant pressure only allow observation of structural effects at very few well-defined compression states. In contrast, the use of the COVASP technique has the huge advantage over classical LB films that it allows the capture of any compression-driven feature, at any pressure, within the same film, which can later be observed in detail, once the different surface pressures in the immobilized film are calibrated [29]. The resulting supported film was observed under an epifluorescence microscope (Leica microsystems, Wetzlar, Germany) equipped with a Hamamatsu digital camera.

## 3. Results

Dynamic light scattering was used to evaluate the possible interaction of PHA nanoparticles with different components of pulmonary surfactant since this technique allows the study of the formation of lipid and protein coronas on nanomaterials [30,31]. To that end, the distribution of particle size in NS, OE, PL, and DPPC multilamellar suspensions was assessed in the absence and presence of NPs (Figure 2a). PHA nanoparticles had a hydrodynamic radius, *R*_H_, of 63 ± 4 nm and a narrow distribution, while the different surfactant preparations exhibited broad heterogeneous distributions. The presence of heterogeneous distributions in lipid and lipid/protein suspensions is not surprising, as it can be frequently seen in the literature even when homogenizing procedures such as sonication or extrusion are used [32,33] as a consequence of the particular behavior of the different components in the mixtures. The different size distributions shown in Figure 2a for NS, OE, and MLVs of DPPC and DPPC/POPG can be explained by the presence of surfactant proteins in NS and OE and the net charge and the potential segregation of phases differing in order/packing and fluidity in the different liposomes. For OE, the presence of surfactant proteins SP-B and SP-C likely promotes the establishment of membrane–membrane contacts that increase the cohesivity between surfactant layers by simultaneous binding to different membranes [2]. This is likely the basis of the monomodal distribution observed. For NS, the binding of SP-A to surfactant lipids and SP-B [34] may somehow prevent the cohesion of surfactant membranes promoted by SP-B and SP-C. This feature could be related with the bimodal distribution observed. Regarding MLVs of DPPC or DPPC/POPG, the absence of surfactant proteins and the subsequent decrease in membrane/membrane interactions could contribute to the observed multimodal size distributions. On the other hand, it has been shown that the presence of unsaturations in lipid acyl chains alters the lamellarity of lipid membranes, increasing the heterogeneity of lipid preparations [32]. Therefore, it is likely that the differing size distributions of DPPC and DPPC/POPG membranes could be due to the C9-10 unsaturation in the oleic chain of POPG. Moreover, the presence of unsaturated species decreases the membrane bending rigidity, favoring self-assembly of the lipids in smaller multilamellar vesicles upon film hydration [32]. Additionally, the negative charges of POPG would also contribute to the different size distributions of DPPC and DPPC/POPG suspensions. Segregation of membrane patches with differences in electrostatic repulsion (i.e., between regions particularly enriched in negatively charged POPG molecules within the DPPC/POPG samples) would also favor the formation of liposomes with different sizes, including a fraction with sizes smaller than those determined for pure DPPC membranes [35].

As large particles diffuse more slowly than small ones [36], the nanoparticle-induced displacement of the correlogram of all surfactant preparations to shorter times (Figure 2b) indicates the formation of new aggregates with a diffusion intermediate between those of the nanomaterial and that of the different surfactant aggregates. This is therefore an indication that surfactant proteins and lipids bind to the nanomaterial surface forming a corona. As a result of this interaction, the peak of nude NPs disappeared while the peaks corresponding to NS, OE, and PL vesicles were shifted towards smaller sizes (Figure 2a). Interestingly, this effect was stronger for NS and PL than for OE. This could indicate that: (i) surfactant protein SP-A, which is present in NS but not in OE, might facilitate the interaction with the nanomaterial and/or (ii) proteins SP-B and SP-C, which are present in OE but not in PL, would prevent the interaction of the nanomaterial with surfactant lipids. For DPPC multilamellar vesicles, the interaction with the nanomaterial induced the disappearance of the nanoparticle peak and shifted the lipid peaks to larger sizes (Figure 2a) (form *R*_H_ = 147 ± 3 nm to 180 ± 5 nm, and from 1010 ± 9 nm to 1932 ± 20 nm). The comparison of NPs effects on the size distributions of PL and DPPC suspensions suggests that NPs have a stronger affinity for POPG than for DPPC.

To gain insight into the interaction of NPs with pulmonary surfactant lipids, we evaluated the effect of the nanomaterial on the compression isotherms of DPPC and PL interfacial films. Figure 3a shows the effect of PHA nanoparticles on the π-A isotherms and the compressibility modulus of DPPC films. In the absence of nanoparticles, the isotherm exhibited a plateau at 9–12 mN/m, indicative of liquid expanded (LE)/tilted condensed (TC) phase coexistence, and a collapse pressure of 70 ± 2 mN/m (Figure 3(a1)). The deposition of low amounts (lipid/nanomaterial weight ratio of 1:0.13) of NPs onto a preformed DPPC monolayer had a negligible effect on the compression isotherm. Increasing the amount of deposited NPs up to a lipid/nanomaterial weight ratio of 1:0.33 shifted the isotherm to lower molecular areas (Figure 3(a1)). This can be attributed to the incorporation of the nanomaterial into the monolayer and the subsequent binding and exclusion of lipid molecules upon the squeeze-out of NPs during film compression. As a result, the monolayer was destabilized and it collapsed at lower surface pressures than in the absence of nanoparticles (64 ± 1 mN/m). A further increase in the amount of deposited NPs (1:0.67 lipid/nanomaterial weight ratio) resulted in a compression isotherm that almost overlapped with that of pure DPPC films, and the collapse pressure partly recovered (67 ± 1 mN/m) (Figure 3(a1)). This indicates that the interaction of NPs with the DPPC monolayer is hindered at small lipid/nanomaterial weight ratios. Compression isotherms were analyzed in terms of the compressibility modulus ($C_{S}^{-1}$). For pure DPPC films, the LE/TC transition appeared as a pronounced minimum of $C_{S}^{-1}$ at a surface pressure of 9 mN/m, which separated the LE phase ($C_{S}^{-1}$ values between 12 and 50 mN/m) and the TC phase ($C_{S}^{-1}$~100–200 mN/m) (Figure 3(a2)). Interaction of the nanomaterial with the DPPC monolayer slightly shifted the minimum to higher surface pressures (10 mN/m), which indicates that NPs stabilized the LE phase. In addition, in the presence of NPs, the maximum value of the compressibility modulus decreased from 158 ± 2 mN/m for pure DPPC films to 144 ± 3 and 146 ± 2 mN/m, for 1:0.33 and 1:0.67 lipid/nanomaterial weight ratios, respectively, indicative of a slight fluidization of the interfacial film.

On the other hand, the deposition of the nanomaterial onto a monolayer composed of DPPC/POPG (7:3, *w*/*w*) had no impact on the qualitative features of the film for all the lipid/nanomaterial weight ratios analyzed (Figure 3(b1)). However, the analysis of the isotherms in terms of the compressibility modulus indicates that large amounts of NPs again stabilized the LE phase and somehow reduced the maximum compressibility value (from 110 ± 2 mN/m for pure PL films to 101 ± 1 and 104 ± 2 mN/m for lipid/nanomaterial weight ratios of 1:0.33 and 1:0.67, respectively), indicative of fluidization of the films (Figure 3(b2)).

The perturbing influence of NPs on DPPC and PL monolayers was further studied by following relaxation kinetics of the lipid films at constant surface pressure for a lipid/NPs weight ratio of 1:0.33. At a surface pressure of 47 mN/m, the deposition of NPs onto a DPPC film did not affect its time-dependent relaxation kinetics, which showed no area loss during the examined relaxation period (Figure 4a). However, deposition of NPs onto a preformed monolayer of DPPC/POPG (7:3, *w*/*w*) subtly destabilized the film, promoting a reduction in the area occupied by lipids in a two-step process (Figure 4a): a slow area reduction during the first 10 min, probably due to the reorganization of lipids upon addition of the nanomaterial, followed by a sharp area decrease indicative of NPs-promoted lipid withdrawal. NPs exerted a more pronounced effect at a surface pressure of 33 mN/m, at which the lipids were less tightly packed, with a more conspicuous NP-promoted reduction in surface area for both lipid films (Figure 4a). It is interesting to note that DPPC and DPPC/POPG films were affected differently by NPs. Namely, deposition of the nanomaterial induced a slight destabilization of the DPPC film for 15 min after which the monolayer became stable; the films containing anionic phospholipids showed a biphasic desorption process similar to that observed at 47 mN/m, although with a steeper slope.

According to the nucleation and growth model of monolayer collapse proposed by Smith and Berg [26], if the molecular loss from the monolayer is due to desorption of lipid molecules into the subphase, a linear relationship between −log(A/A_0_) and the square root of time should be obtained, whereas the absence of a linear relationship would indicate that the area loss during relaxation would be caused by the formation of three-dimensional aggregates. Figure 4b shows that, at a surface pressure of 47 mN/m, a linear relationship between −log(A/A_0_) and $\sqrt{t}$ was obtained, where A is the monolayer area at a given time and A_0_ the area at t = 0 min. This indicates that the monolayer molecular loss is due to NP-induced desorption of lipid molecules into the subphase. The desorption rate values obtained for both lipid films at 47 and 33 mN/m (Table 1) and the small changes observed in the A/A_0_ ratios (Figure 4a) indicate that the process occurs slowly and that only trace amounts of lipids would be withdrawn from the interfacial films.

To determine whether the effect of NPs on phospholipid monolayers could be due to the incorporation of some PHA molecules out from the NPs within the interfacial film, we evaluated the effect of the presence of PHA on the compression isotherms of DPPC and PL films formed already in the presence of bioplastic. To that end, PHA, dissolved in acetone, was mixed with the lipids in chloroform/methanol and the lipid/PHA mixture was deposited onto the air/liquid interface, forming a mixed lipid/PHA monolayer. In contrast to what was observed for DPPC monolayers in the presence of NPs, the compression isotherms of DPPC films shifted to higher molecular areas in the presence of PHA molecules (lipid/PHA weight ratio of 1:0.33) (Figure 5(a1)), which indicates that PHA molecules incorporated and occupied some space at the interfacial film. It is interesting to note that this effect was partly abolished at larger amounts of PHA for which a shorter displacement of the isotherm was observed (Figure 5(a1)). This would indicate that incorporation of PHA molecules into the monolayer is hindered at high PHA content. Likely, the deposition of large amounts of PHA on the air/liquid interface and the subsequent diffusion of acetone away from the plastic into the subphase, or its evaporation towards the air side, might induce the rapid formation of polymer aggregates that could translocate into the subphase largely without affecting the lipid film. Evaluation of the compressibility modulus of DPPC films in the absence and presence of PHA (Figure 5(a2)) indicates that the incorporation of PHA molecules fluidized the film, shifting the LE/TC phase coexistence to higher surface pressures (from 9 mN/m in the absence of PHA to 11 mN/m in the presence of PHA) and decreasing the compressibility moduli of the TC phase. On the other hand, PHA molecules at a lipid/PHA weight ratio of 1:0.33 also destabilized PL films, as indicated by the shift of the compression isotherm to lower molecular areas and the decrease in collapse pressure (from 70 ± 1 mN/m to 65 ± 1 mN/m), indicative of PHA-promoted lipid sequestration. Increasing PHA content to a lipid/polymer weight ratio of 1:0.67 resulted in minor changes in the compression isotherm of PL, which slightly shifted to lower molecular areas (Figure 5(b1)). The interaction of PHA with PL films decreased the compressibility moduli of the TC phase without affecting the LE phase (Figure 5(b2)). This effect was weaker for the largest content of PHA assayed, which would indicate that incorporation of large amounts of PHA in the PL monolayer is also hampered. Taken together, our results indicate that NP-induced destabilization of DPPC and PL films would be due to the direct binding of lipid molecules to the nanomaterial surface and not to the transfer of PHA molecules into the monolayers.

To characterize the impact of NPs and PHA on the LE/TC phase transition behavior of surfactant phospholipid films subjected to compression, both DPPC and DPPC/POPG lipid films, doped with the fluorescent lipid NBD-PC, were transferred onto a solid support and visualized under epifluorescence microscopy in the absence and presence of NPs or PHA. Pure DPPC films segregated under compression dark multilobed domains characteristic of the TC phase, which grew in size as the surface pressure increased (Figure 6). The comparison of these images with those obtained in the presence of NPs illustrates how NPs decreased the size and number of condensed domains (Figure 6), causing a relative decrease of the LC phase and the subsequent fluidization of the monolayer. NPs also altered the morphology of the condensed domains, which became dendritic and were encircled by a gray halo. Given that the fluorescent probe partitions into the LE phase and is excluded from the more tightly packed TC phase, the gray-like phase would correspond to a new lipid phase, with a lipid packing intermediate between those of the LE and TC phases, in which the fluorescent probe incorporated to a lesser extent than in the LE phase. On the other hand, the incorporation of PHA molecules into DPPC films induced the complete disappearance of the LE phase and the appearance of the gray phase (Figure 6). In addition, PHA facilitated the nucleation of condensed domains but hampered their growth (Figure 6), fluidizing the monolayer in agreement with the reduction in the compressibility modulus of the DPPC/PHA isotherm described above. For surface pressures above 11 mN/m, black solid domains formed reticular structures enclosed by the gray phase and bright spots of LE phase appeared (Figure 6), suggesting a preferential interaction of PHA molecules with the fluid/condensed domain boundaries.

In the case of the anionic DPPC/POPG PL films, LE/TC phase coexistence was observed for all the surface pressures assayed (Figure 7). Deposition of the nanomaterial onto preformed PL films stabilized the LE phase, shifting the LE/TC phase coexistence to higher surface pressures. Since PL compression isotherms were not affected by NPs, it is conceivable that small TC domains could still be formed, whose sizes could perhaps be below the optical resolution of a light microscope. This hypothesis is reinforced by the finding that the TC domains observed in the presence of NPs were smaller and more numerous than those observed in the absence of the nanomaterial at any given surface pressure (Figure 7), as described above for pure DPPC films. Moreover, NPs also induced the appearance of the probe-excluding gray phase, which distributed evenly in the film when TC domains appeared and formed reticular structures enclosing condensed domains as the surface pressure increased. On the other hand, the incorporation of PHA molecules into the PL monolayer exerted a further fluidizing effect, increasing the surface pressure at which phase coexistence appeared and decreasing the size of condensed domains (Figure 7). These results agree with the observed effect of PHA on PL compression isotherms. PHA also induced the appearance of the gray phase, which distributed randomly within the monolayer at all the surface pressures assayed.

## 4. Discussion

Pulmonary surfactant constitutes the first barrier that inhaled particles meet in the alveolar region. Different studies have demonstrated that surfactant components interact with nanoparticles. These interactions can determine the lifetime, fate, and toxicity of the nanomaterial in the airways [9,37] as well as adversely affect pulmonary surfactant function [6]. Therefore, in this work, we evaluated whether PHA nanoparticles are suitable for lung drug delivery by studying the interaction of this nanomaterial with pulmonary surfactant and interfacial films made of the main surfactant lipids.

Our results demonstrate that NPs interact with surfactant proteins and lipids. In this regard, DLS measurements of NPs in the presence of native pulmonary surfactant or suspensions made of the organic extract of pulmonary surfactant, a mixture of DPPC/POPG or pure DPPC multilamellar vesicles, show that the nanomaterial interacts with surfactant lipids and that this interaction could be promoted by SP-A, present in our natural surfactant preparation. As a matter of fact, a role of SP-A promoting surfactant lipid binding to different nanomaterials has been reported [38]. This effect may be related to the oligomeric nature of SP-A and its ability to bind to different ligands through diverse domains, which would facilitate the simultaneous binding of the protein to the nanomaterial and to surfactant lipids. In contrast, SP-B and SP-C did not facilitate the interaction of surfactant lipids with NPs. These results correlate with those of Wohlleben and coworkers [38], who found that SP-B only promotes lipid binding to nanoparticles functionalized with amino or PEG residues.

Regarding the interaction of surfactant lipids with PHA nanoparticles, we demonstrated that NPs interact with vesicles and films composed of DPPC or a mixture of DPPC/POPG 7:3 (*w*/*w*). It is noteworthy that NPs did not substantially affect the compression isotherms or the compressibility of both types of lipid films at lipid/nanomaterial weight ratios below 1:0.33. However, at NPs concentrations ≥ 1:0.33, a fluidizing effect was observed in both monolayers, this effect being weaker for a lipid/NP weight ratio of 1:0.67 than for 1:0.33. This suggests that the nanomaterial may agglomerate and perhaps segregate at the interface, which would contribute to the loss of NPs to the subphase. This hypothesis is supported by some studies that show that nanoparticles spread onto a preformed DPPC monolayer agglomerate upon solvent evaporation [39,40,41]. Increasing the concentration of deposited nanomaterial would increase agglomerate size, facilitating its loss towards the subphase. Loss of nanomaterial to the subphase was also indicated by NPs effects on both DPPC and PL films since penetration of NPs into the monolayers would shift the compression isotherms to higher molecular areas. Thus, the shifting of the DPPC isotherm to lower molecular areas would indicate that NPs could somehow sequester DPPC molecules once translocating to the subphase [41,42]. On the other hand, the finding that isotherms of PL films following deposition of NPs were identical to the control suggests that NPs do not remain associated but translocate to the subphase. Because of the hydrophobic nature of PHA, one could expect that NPs would remain associated to the interfacial film instead of translocating to the subphase [43]. However, the hydrophobic interactions between lipid acyl chains and the nanomaterial surface would promote the formation of a lipid corona in which the lipid polar headgroups would increase the hydrophilicity of the nanoparticle/lipid complex, facilitating the migration of the nanomaterial towards the aqueous subphase [44]. In this regard, the relaxation experiments performed at constant surface pressure demonstrated that NPs promoted a slight desorption of lipid molecules from both DPPC and anionic PL films, this effect being stronger for the latter.

NPs also incorporated into both tested phospholipid films, leading to the appearance of perturbed regions appearing as a gray-like phase under the fluorescence microscope. These regions could represent a sort of “alloy” in which the presence of bioplastic could facilitate the intermixing of TC and LE nanodomains. This would partly prevent the incorporation of the fluorescent probe, at least with the same density as observed in the pure LE phase, resulting in a less bright region. For pure DPPC films, dark TC domains surrounded by that gray phase in a bright LE background were observed, indicative of a clear preference of the nanomaterial for defect structures at the fluid-condensed boundaries. However, for anionic monolayers, NPs apparently distributed evenly into the monolayer, both in the fluid POPG-enriched phase and at LE/TC phase coexistence boundaries. The location of NPs at the LE/TC borders would decrease the line tension imposing an energy barrier to domain coalescence, kinetically stabilizing them against merging, hence decreasing the size of condensed domains [45]. Since domain growth requires the migration of lipid molecules to the closest nuclei [45], the increased lipid packing of the NPs-promoted gray phase would increase monolayer viscosity [46], decreasing the rate of domain motion, thus hindering their growth [45,47]. This behavior recalls that of cholesterol, which has been shown to slow down domain coarsening in surfactant monolayers by affecting the diffusivity of DPPC molecules [48].

An interesting conclusion from our experiments on the interaction of NPs with both bilayers (as observed by DLS) and monolayers (in the Langmuir and epifluorescence experiments) made of the main surfactant phospholipids is that the segregation of ordered/disordered-fluid phases, intrinsic to the typical composition of saturated/unsaturated surfactant phospholipids, may have an impact on the interaction with nanostructured materials. NPs seem to somehow interact selectively with fluid areas, accumulating at boundaries between ordered/disordered regions. Those regions could be particularly favorable to bend at the nanoscale, facilitating the transfer of lipid layers to form lipid- or lipid/protein-based surface coronas. This, on one hand, could substantially influence the fate of NPs through their interactions with subjacent biological systems, i.e., the respiratory epithelium. It may also have some impact on the organization and performance of pulmonary surfactant structures and their fundamental role to stabilize the delicate structure of alveoli against the demanding breathing mechanics.

Still, and despite the reduction observed in the size of condensed domains, the tested lipid films retained their ability to reduce surface tension upon compression to very low values in the presence of NPs. This suggests that the formation of a network of the nanomaterial-induced phase in which condensed domains are embedded would maintain the properties of a continuous condensed phase, with the stability and flexibility required to attain and support high surface pressures (low surface tensions) upon compression. The fact that NPs slightly decreased the compressibility modulus of the TC phase in both DPPC and DPPC/POPG films reinforces this hypothesis. Formation of such alloy-like structures is also supported by the observation of nanometer-sized domains in DPPC and DPPC/DPPG films [49] as well as in rat [50] and bovine surfactant monolayers [51]. Furthermore, the formation of networks of DPPC nanodomains has been proposed by mesoscopic kinetic modeling [52].

Our results show that NPs effects on DPPC and PL films were not caused by the transfer of PHA molecules into the interfacial layers. Unlike NPs, PHA incorporated into DPPC films altered van der Waals interactions between adjacent lipid molecules with the consequent fluidization of the monolayer. PHA also destabilized PL films by withdrawing lipid molecules, which decreased the attainable collapse pressure. On the other hand, as previously described for NPs, the interaction of PHA with DPPC and POPG acyl chains promoted the formation of a perturbed phase with a lipid packing intermediate between those of the LE and TC phases. However, the stability that this new phase would confer to both films was not enough to counterbalance the destabilizing effect of PHA, especially in anionic monolayers. Thus, it is likely that the reduced surface area of NPs compared to that of PHA molecules would minimize hydrophobic interactions with surfactant lipids, decreasing the amount of lipids that were lost upon the translocation of NPs towards the subphase.

In addition to the interfacial layer, pulmonary surfactant is composed of a complex three-dimensional network of interconnected membranes that acts as a reservoir of fresh material, ensuring correct recycling of the surfactant structures that have lost their functional properties because of the continuous respiratory cycling and high oxidative stress at the alveolar space [1]. Our results suggest that PHA nanoparticles would cross the surfactant monolayer reaching the alveolar lining fluid. The interaction of the nanomaterial with surfactant films at the air/liquid interface and with the surfactant membranes in the alveolar fluid would result in the formation of a lipoprotein corona. This corona has been shown to critically modulate the toxicity and biodistribution of the nanomaterial [53]. In this regard, it has been reported that such lipoprotein coating promotes the phagocytosis of the nanomaterial by alveolar macrophages [31,54,55] and its eventual uptake by epithelial cells [10] where the nanomaterial can be hydrolyzed by lysosomal lipases [56]. As a result, the intracellular delivery of the cargo would occur, reducing the dose needed to obtain the desired therapeutic effects and the unwanted side effects [57].

## 5. Conclusions

In summary, our results demonstrate that PHA nanoparticles could be used for pulmonary drug delivery since (i) the nanomaterial translocated through the interfacial monolayer into the aqueous phase, which would allow the delivery of the cargo to its target site; and (ii) the interaction of monolayer-incorporated NPs with surfactant lipids promoted the formation of a new lipid phase with characteristics intermediate between those of the LE and TC phases that still allowed the attainment of high surface pressures (very low surface tensions) upon compression. Further studies to consider the role of surfactant proteins on the interaction of NPs with surfactant membranes are guaranteed to confirm the feasibility of PHA nanoparticles as pulmonary drug delivery systems.

## Figures and Tables

**Figure 1 nanomaterials-11-01482-f001:**
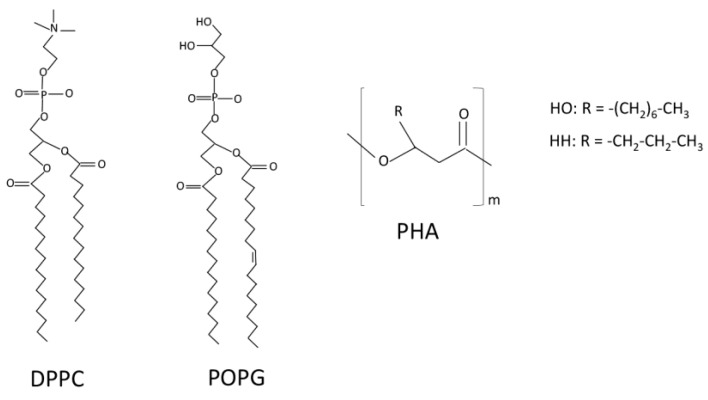
Chemical structure of PHA, DPPC, and POPG. PHA is a copolymer of hydroxyoctanoate (HO) and hydroxyhexanoate (HH).

**Figure 2 nanomaterials-11-01482-f002:**
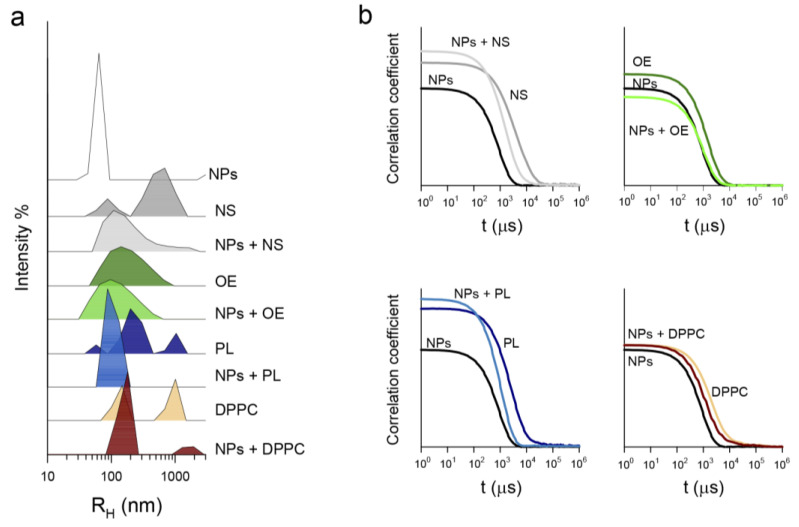
Interaction of PHA nanoparticles with pulmonary surfactant-related materials as determined by DLS. (**a**) Size distributions and (**b**) correlograms. Materials tested in the absence or presence of PHA NPs included NS (native surfactant); OE (the reconstituted organic extract of pulmonary surfactant); PL, a suspension of the lipid mixture DPPC/POPG 7:3 (*w*/*w*); and DPPC multilamellar vesicles. DLS measurements were performed at 25.0 ± 0.1 °C and a lipid/nanomaterial weight ratio of 1:0.33.

**Figure 3 nanomaterials-11-01482-f003:**
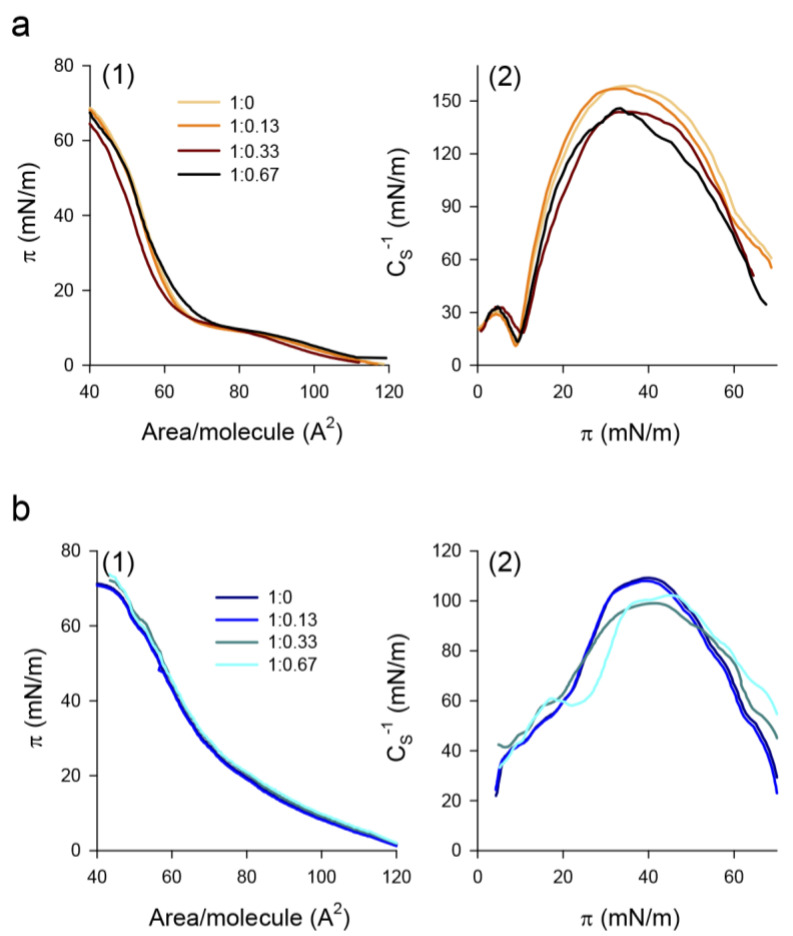
Effect of NPs on compression isotherms and compressibility modulus of interfacial phospholipid films. (**a**) DPPC, (**b**) DPPC/POPG (7:3, *w*/*w*) monolayers. The nanomaterial was deposited onto the preformed monolayer and allowed to interact for 10 min before starting monolayer compression at 50 cm^2^/min. Measurements were performed at 25.0 ± 0.1 °C.

**Figure 4 nanomaterials-11-01482-f004:**
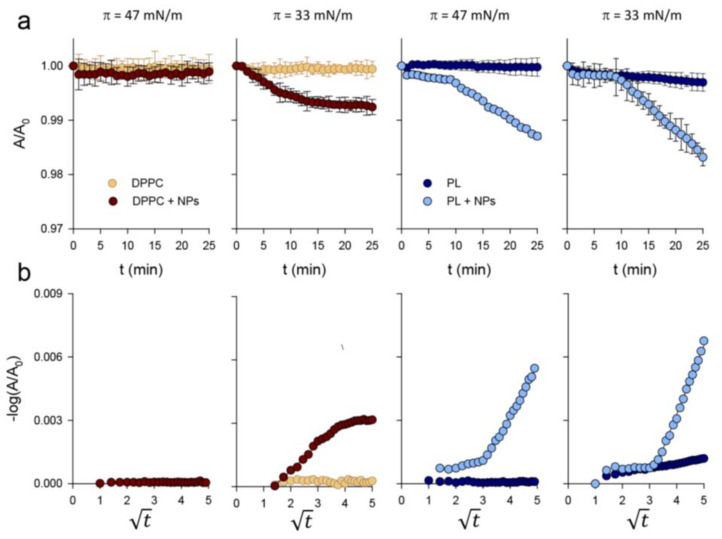
Relaxation of interfacial phospholipid films in the presence of NPs. (**a**) Relaxation kinetics at a constant surface pressure of 47 or 33 mN/m in DPPC and DPPC/POPG (7:3, *w*/*w*) films, in the absence and presence of NPs (lipid/nanomaterial weight ratio of 1:0.33). A and A_0_ are the trough surface areas at a given time, t, and at t = 0 min, respectively. (**b**) Effect of NPs on constant pressure relaxation data for DPPC and PL films expressed as −log(A/A_0_) versus $\sqrt{t}$. Data shown are the mean of three independent measurements. The standard deviation for each relaxation kinetic was too small to be displayed by error bars.

**Figure 5 nanomaterials-11-01482-f005:**
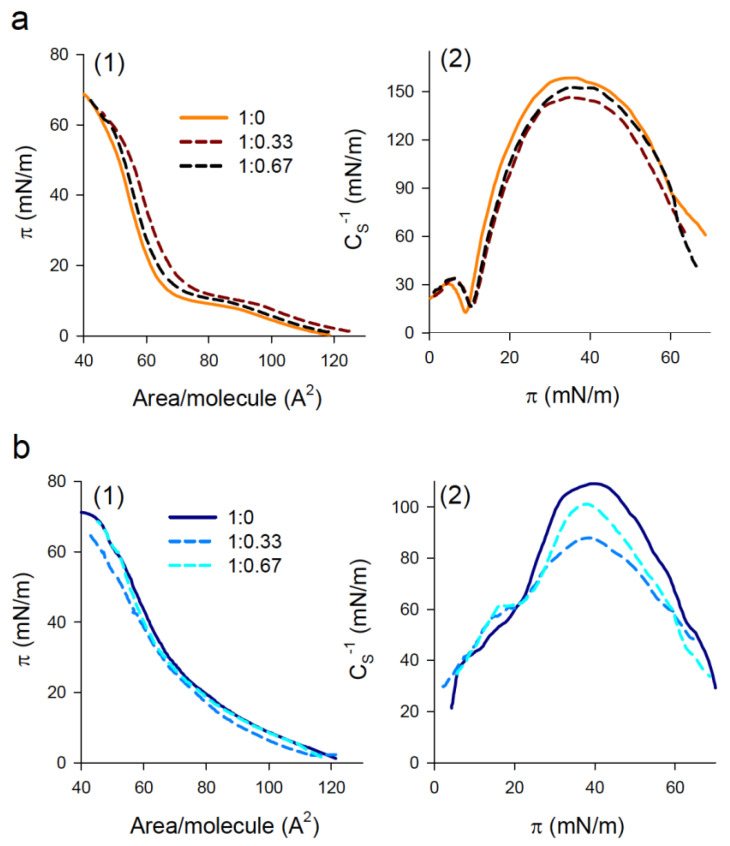
Effect of PHA on the compression isotherms and compressibility modulus of interfacial films. (**a**) DPPC, (**b**) PL films. Mixed PHA/lipid films were obtained by spreading a mixture of PHA, dissolved in acetone, and DPPC or PL, dissolved in chloroform/methanol, onto the air/liquid interface. Measurements were performed at 25.0 ± 0.1 °C and at a compression rate of 50 cm^2^/min.

**Figure 6 nanomaterials-11-01482-f006:**
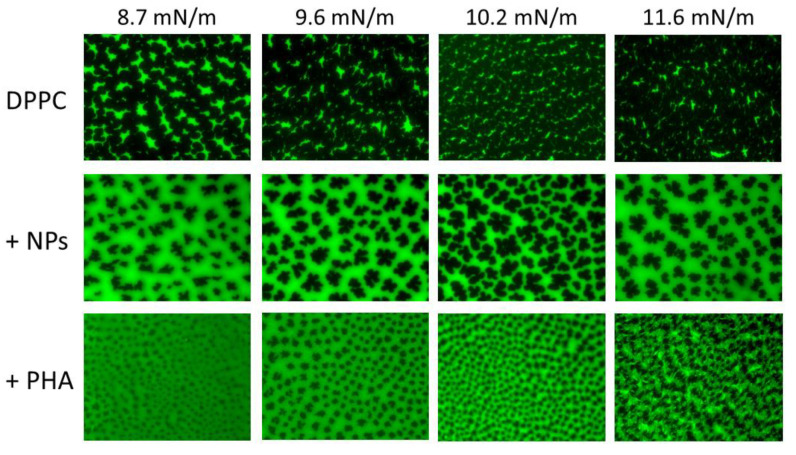
Epifluorescence microscopy of interfacial DPPC films in the absence and presence of NPs and PHA. Images of DPPC, alone and in the presence of NPs or PHA molecules (lipid/nanomaterial or PHA ratio of 1:0.33), at different surface pressures were obtained upon transfer of the corresponding interfacial films onto glass slides and observation under an epifluorescence microscope. DPPC films were doped with NBD-PC, which partitions preferentially into the LE phase.

**Figure 7 nanomaterials-11-01482-f007:**
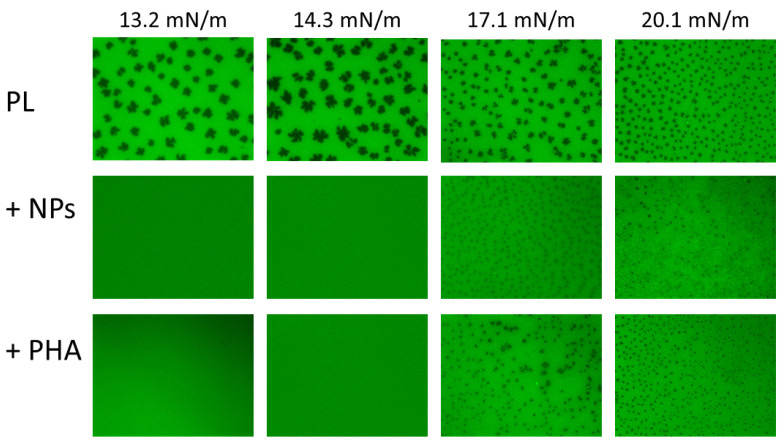
Epifluorescence microscopy of interfacial DPPC/POPG films in the absence and presence of NPs and PHA. Epifluorescence microscopy images of monolayers made of DPPC/POPG (7:3, *w*/*w*), alone and in the presence of NPs or PHA molecules (lipid/nanomaterial or PHA ratio of 1:0.33), at different surface pressures. PL films were doped with NBD-PC, which incorporates in the LE phase.

**Table 1 nanomaterials-11-01482-t001:** Rate desorption constants (*k*_1_, *k*_2_) for the NPs-induced dissolution of DPPC and PL (DPPC/POPG 7:3, *w*/*w*) monolayers compressed at 47 and 33 mN/m.

π (mN/m)	Sample	k_1_ $\left(1/\sqrt{min})\;(r^{2}\right)$	k_2_ $\left(1/\sqrt{min})\;(r^{2}\right)$
**47**	DPPC	-	-
	DPPC + NPs	-	-
	PL	-	-
	PL + NPs	0.0002 (0.992)	0.002 (0.993)
**33**	DPPC	-	-
	DPPC + NPs	0.001 (0.997)	-
	PL	0.0002 (0.993)	-
	PL + NPs	0.003 (0.995)	0.003 (0.997)

The goodness of fit is given by the linear regression coefficient (*r*^2^).

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
