# Peer review of "Polyhydroxyalkanoate Nanoparticles for Pulmonary Drug Delivery: Interaction with Lung Surfactant"

_nanomaterials, 2021, doi:10.3390/nano11061482_

Round 1

Reviewer 1 Report

The paper for Nanomaterials edited by Canadas et al. describes the interaction of polyhydroxyalkanoate nanoparticles (PHA) with the lung surfactant biophase. The study, conducted using different and complementary investigation techniques (e.g. DLS, isotherms analysis and in vitro experiments), deals with an interesting topic, producing the knowledge for application in drug delivery of the proposed nanoparticles. Using different surfactants and organic extracts, lung models are proposed and examined.

The paper is generally well-written. The readability is good and the paper has scientific coherence. In my opinion, the manuscript is suitable for publication after minor revision, listed following

  • To improve the clarity, I suggest to include in the introduction a figure with the chemical formulas of all the components.
  • Reference 8 seems missing. Please, check this point.
  • DLS intensities are not useful for analysis reported in Figure 1. Please, include the correlation functions. In addition, NPs seems to be 100 nm not 63 are reported in text.

Author Response

Comments and Suggestions for Authors

The paper for Nanomaterials edited by Canadas et al. describes the interaction of polyhydroxyalkanoate nanoparticles (PHA) with the lung surfactant biophase. The study, conducted using different and complementary investigation techniques (e.g. DLS, isotherms analysis and in vitro experiments), deals with an interesting topic, producing the knowledge for application in drug delivery of the proposed nanoparticles. Using different surfactants and organic extracts, lung models are proposed and examined.

The paper is generally well-written. The readability is good and the paper has scientific coherence. In my opinion, the manuscript is suitable for publication after minor revision, listed following

  • To improve the clarity, I suggest to include in the introduction a figure with the chemical formulas of all the components.

Response: According to the suggestion, we have included a new figure, now Figure 1, illustrating the chemical structures of DPPC, POPG and PHA.

  • Reference 8 seems missing. Please, check this point.

Response: Instead of the previous reference 8, we have included in the revised version of the manuscript two references of articles illustrating how the exposure of nanoparticles to surfactant can certainly alter their properties and toxicity:

Sweeney, S.; Leo, B.F.; Chen, S.; Abraham-Thomas, N.; Thorley, A.J.; Gow, A.; Schwander, S.; Zhang, J.J.; Shaffer, M.S.P.; Chung, K.F.; Ryan, M.P.; Porter, A.E.; Tetley, T.D. Pulmonary surfactant mitigates silver nanoparticle toxicity in human alveolar type-I-like epithelial cells. Colloids Surf. B Biointerfaces. 2016, 145, 167-175.

Radiom, M.; Sarkis, M.; Brookes, O.; Oikonomou, E.K.; Baeza-Squiban, A.; Berret, J.F. Pulmonary surfactant inhibition of nanoparticle uptake by alveolar epithelial cells. Sci. Rep. 2020, 10, 19436.

  • DLS intensities are not useful for analysis reported in Figure 1. Please, include the correlation functions. In addition, NPs seems to be 100 nm not 63 are reported in text.

Response: In line with the referee’s suggestion, we have now included the correlation functions in the figure (now Figure 2, in page 6) and discussed them in the new version of the manuscript (pages 5 and 6). We thank the reviewer for noticing the mistake in the plot of the size distribution of PHA nanoparticles. It has been corrected in the new version of the figure.

Reviewer 2 Report

The contribution by O. Cañadas and co-authors regards the interactions of the polyhydroxyalkanoate nanoparticles with models of lung surfactants. As models multilamellar vesicles (MLV), Langmuir monolayers, and monolayers on solid support are applied. The studies of the interactions of possible drug carriers on lung surfactant, especially now in the pandemics of Covid-19, are of utmost importance. However, any model studies should be reasonably planned and performed following the usually well established rules of the art in a given experimental subdiscipline. Here I am very skeptical regarding the means of Langmuir monolayers’ formation and their further investigation. Therefore, I do not recommend the publication of this contribution.

Comment 1

P4 l-160-164

“To evaluate the effect of NPs on π-area isotherms, different  volumes of an aqueous suspension of the nanomaterial were deposited onto the preformed monolayer and allowed to interact with the interfacial film for 10 min. Then, compression isotherms were recorded. Alternatively, PHA, dissolved in acetone, was added to the lipid organic solution and the mixture was deposited onto the air-liquid interface.”

I am skeptical regarding this experimental procedure. What does it mean that “different volumes of an aqueous suspension of the nanomaterial were deposited onto the preformed monolayer”? I understand that the chloroform/methanol solution of DPPC or DPPC/POPG mixture was first deposited at the tris buffer/air interface and then the suspension of the PHA nanoparticles was deposited in different places of the Langmuir trough area by a microsyringe? The problem is that the “preformed monolayer” at 0 surface pressure is in the gas(G)/liquid expanded (LE) state equilibrium, so statistically pure buffer dominates at the interface. Thus, most of the NPs can be suspended in the bulk subphase and their interaction with the monolayer can be negligible, which is later on proved by the π-A isotherms presented in Fig.2.

A much serious problem is connected with the second procedure. Acetone solution of PHA was mixed with the chloroform/methanol solution of DPPC (DPPC/POPG). The first question here regards the mutual proportion of acetone and the other solvents in the final spreading solution. What was the proportion, was it constant? Was a blank experiment performed? It is well known from the very beginning of the experiments on Langmuir monolayers (times of I. Langmuir and W. Harkins) that the course of the π-A isotherm is affected by the composition of the spreading solution. Thus, the effects illustrated in Fig. 4. were caused in my opinion by the addition of acetone and not by the presence of PHA in the solution.

Comment 2

P4 l 189-191

“Afterwards, the interfacial film was subjected to compression at a constant speed of 25 cm2/min, while transferring the interface onto a glass coverslip that had been previously immersed into the subphase (27).”

Reading this sentence I understand that the authors transferred the monolayer from the tris/air interface on the glass support applying the Langmuir-Blodgett technique. However in this technique the monolayer is first compressed to a given constant surface pressure value, stabilized at this conditions and then the deposition is performed. Reading the above description I have the notion that the deposition was performed simultaneously with the monolayer compression.

Comment 3

P 5 Figure 1

Please check and correct the abbreviation of one of the systems: should it be EO or OE?

Comment 4

In their experiment the authors hydrated deposited lipid films producing multilayer liposomes (MLV). As it can be seen in Fig. 1. the distribution of RH in NS was bimodal, in OE monomodal, in PL (DPPC/POPG mixture) trimodal and in DPPC bimodal. This is a very strange result which was not commented by the authors. Especially I do not completely understand the trimodal distribution of the MLV in the DPPC/POPG (0,7/0,3) mixture or the bimodal distribution of the DPPC MLV. Such a distribution for a simple monolipid system (DPPC) or a binary system (DPPC/POPG) is nonphysical.

Comment 5

P6 Figure 2

This figure is illegible. Why do the authors use the line+symbol template? I guess that in the course of the pi-A isotherm hundreds of data points were registered (it depends on the settings of the NIMA software). Thus, please remove the symbols and use only solid lines in both the isotherms and modulus-π dependences.

Comment 6

Interpretation of Fig. 2.

I do not agree with the interpretation provided by the authors. The typical experimental uncertainty for mean molecular area for NIMA or KSV-NIMA trough is +- 1 Å2. The isotherms shown in Fig. 2. overlap within the experimental error. The only curve which is slightly shifted to lower mean molecular areas is this designed 1:0.33. However, the explanation given by the authors is illogical: “This indicates that the interaction of NPs with the DPPC monolayer is hindered at small lipid/nanomaterial weight ratios.” I completely do not agree. If this shift is caused by the desorption of some DPPC molecules from the interface to the bulk subphase by the NPs the rising number of the NPs at the interface should increase this effect. In my opinion the slight shift observed for the 1:0.33 curve has a random character and the investigated NPs do not affect the monolayer characteristics, as they are not present at the interface due to experimental reasons mentioned in comment 1. Moreover, the changes in the value of compression modulus are also minor and their interpretation does not add new quality to the interpretation of the results.

Comment 7

Fig3. Row A panel 3 – please add the error bars to the set of points lacking it.

Comment 8

P8 l 296-298 and Fig. 4.

“To that end, PHA, dissolved in acetone, was mixed with the lipids in chloroform/methanol and the lipid/PHA mixture was deposited onto the air/liquid interface, forming a mixed lipid/PHA monolayer.”

In my opinion all the changes in the courses of π-A isotherms presented in Fig. 4. Are caused by the addition of acetone to the spreading solution and not by the incorporation of the polymer molecules to the monolayer. Any real incorporation of the polymer to the monolayer would profoundly change the course of the isotherm and especially would cause a dramatic increase of the A0 area (beginning of surface pressure rise). Nothing of this was observed. Moreover, this time the 1:0.33 curve is shifted to larger mean molecular areas which is in contradiction with the results reported in Fig. 2. Moreover, if any incorporation took place the 1:0.67 curve would be shifted to larger mean molecular areas than this designed 1:0.33 or at leas overlap it (assuming limited incorporation of the polymer molecules).

Comment 9

P 11 l 339

“Given that the fluorescent probe partitions into the LE phase and is excluded from the more tightly packed TC phase, the gray-like phase would correspond to a new lipid phase, with a lipid packing intermediate between those of the LE and TC. phases, in which the fluorescent probe incorporated to a lesser extent than in the LE phase. On the other hand, incorporation of PHA molecules into DPPC films induced the complete disappearance of the LE phase and the appearance of the gray phase (Figure 5).”

In my opinion the presence of the “gray phase” in the photos of the fluorescent microscope is connected with some experimental problems and not with the physics of the DPPC monolayer. The phases in DPPC and other phospholipid molecules are well described in the scientific literature. The LE phase means molecules with melted chains (some equivalent of the Ld phase in bilayers) without any correlation or order of the long axes of the hydrocarbon chains. The liquid condensed (LC) phase called here tilted condensed (TC) is a 2D crystalline phase with the hydrocarbon chains collectively tilted from the normal. DPPC as a phosphatidylcholine has a large headgroup, so the tilt of its hydrocarbon chains is large exceeding 30 deg even at high surface pressures (as 33 mN/m in the studies). There is no place for an additional “mesophase” between LE and LC. The chains are either collectively ordered or disordered.

Comment 10.

In my opinion the presence of NPs within the lipid monolayer should be directly proved. As the NPs are smaller than the resolution of any optical microscope, their presence and distribution should be documented by different techniques. In my opinion the technique of first choice is here atomic force microscopy. The experiments performed by the authors proved that it was relatively easy to transfer the monolayers on solid supports, so the further step should be the thorough analysis of their texture with the application of AFM.

Author Response

Please, se the attachment

Reviewer 3 Report

The article present an interesting  study reagarding the feasibiity of the use of nanoparticles. 

It is clear that the particles arise the point in longes where should deliver the drug encpasulated. However, to me it is not well described and studied how the delivery wll take place. Will be the particles destilize by any extenal agent?

Author Response

Comments and Suggestions for Authors

The article present an interesting  study regarding the feasibility of the use of nanoparticles. 

It is clear that the particles arise the point in longes where should deliver the drug encapsulated. However, to me it is not well described and studied how the delivery will take place. Will be the particles destilized by any external agent?

Response: PHA polymers are typically degraded by the action of nonspecific lipases and esterases[1]. Therefore, it is likely that PHA nanoparticles reaching the alveolar fluid could end somehow hydrolyzed by phospholipases and esterases that are present in the medium, facilitating the progressive delivery of the cargo. In this regard, it has been shown that the activity of phospholipase A2 is increased in certain pulmonary pathologies like acute respiratory distress syndrome (ARDS)[2], which would facilitate the degradation of the nanomaterial.

Moreover, surfactant proteins and lipids have been shown to form a corona on the surface of different nanoassemblies[3]. Such lipoprotein coating facilitates the phagocytosis of the nanomaterial by alveolar macrophages[4] and its eventual uptake by respiratory epithelial cells[5]. Hence, once delivered at the lysosomes, the degradation of the nanoparticles by lysosomal lipases could also end in the intracellular delivery of the cargo.

We have included a small paragraph at the end of the Discussion (in page 14) for the readers to envision the potential advantage of using these bioplastic nanoparticles in inhalative drug delivery.

[1] Mukai K, Doi Y, Sema Y, Tomita K. 1993. Substrate specificities in hydrolysis of polyhydroxyalkanoates by microbial esterases. Biotechnol. Lett. 15: 601-604.

[2] Kim DK, Fukuda T, Thompson BT, Cockrill B, Bonventre JV. 1995. Bronchoalveolar lavage fluid phospholipase A2 activities are increased in human adult respiratory distress syndrome. Am J Physiol Lung Cell Mol Physiol. 269: L109-L118.

[3] Hu Q, Bai X, Hu G, Zuo YY. 2017. Unveiling the molecular structure of pulmonary surfactant corona on nanoparticles. ACS Nano 7: 6832–6842; Raesch S, Tenzer S, Storck W, Rurainski A, Selzer D, Ruge CA, Pérez-Gil J, Schaefer UF, Lehr C-M. Proteomic and lipidomic analysis of nanoparticle corona upon contact with lung surfactant reveals differences in protein, but not lipid composition. ACS Nano. 9: 11872-85.

[4]Ruge CA, Kirch J, Cañadas O, Schneider M, Perez-Gil J, Schaefer UF, Casals C, Lehr CM. 2011. Uptake of nanoparticles by alveolar macrophages is triggered by surfactant protein A. NanomedicineNanotechnologyBiology and Medicine 7:690-693; Ruge CA, Schaefer UF, Herrmann J, Kirch J, Cañadas O, Echaide M, Pérez-Gil J, Casals C, Müller R, Lehr CM. 2012. The interplay of lung surfactant proteins and lipids assimilates the macrophage clearance of nanoparticles. PLoS One. 7:e40775; Ruge C, Hillaireau H, Grabowski N, Beck-Broichsitter M, Cañadas O, Tsapis N, Casals C, Nicolas J, Fattal E. 2016. Pulmonary surfactant protein A – mediated enrichment of surface-decorated polymeric nanoparticles in alveolar macrophages. Mol. Pharmaceutics. 13: 4168-4178.

[5] Merckx, P.; De Backer, L.; Van Hoecke, L.; Guagliardo, R.; Echaide, M.; Baatsen, P.; Olmeda, B.; Saelens, X.; Pérez-Gil, J.; De Smedt, S.; Raemdonck, K.2018.  Surfactant protein B (SP-B) enhances the cellular siRNA delivery of proteolipid coated nanogels for inhalation therapy. Acta Biomater., 78:236-246.

Round 2

Reviewer 2 Report

After the review process the authors introduced some corrections into the original text; however, in my opinion these changes have only a cosmetic character; and all the problems indicated in my first review are still present in the text. I think that the paper is not publishable in the present form; however, in my opinion the reconstruction of the experimental part, re-writing it in an more precise and clear way, explanation of all the initial ideas and assumptions would make the article finally publishable.

The authors provided long, partially convincing responses to my remarks from the first review. However, more of these explanations and ideas should be applied by the authors in the in-depth discussion of their experimental results in the manuscript.

Experimental

Regarding the preparation of the monolayers:

There are two approaches: in one of them the suspension of PHA is deposited on top of the preformed monolayer. In the second approach the acetone solution of PHA is first mixed with the chloroform/methanol solution of DPPC or DPPC/POPG mixture and then deposited at the buffer/air interface. In the experimental section the description of the first approach its incomplete, whereas the second approach is not described at all.

Approach 1

“different volumes of PHA nanoparticles suspensions were deposited by a microsyringe at different places of the Langmuir trough and allowed to interact with the interfacial film for 10 min.”

Further on in the text (description of Fig. 3. P6, l 232) “Deposition of low amounts (13% by weight) of NPs”.

Following this it is obvious that the authors had a fixed idea regarding the amounts of the deposited NPs. However, this idea is hidden to the readers. In the experimental part regarding the approach 1 the authors should explain which was their idea, how they changed the amounts of the deposited NPs. What do the authors understand also by the term “13% by weight” – how was the weight of the deposited NPs correlated with the weight of the phospholipid molecules forming the Langmuir monolayer. Without a clear and thorough explanation of this questions the paper is very difficult to follow. Moreover, the interactions of Langmuir monolayers with NPs are frequently studied; however, usually the NPs are suspended in the subphase. The approach of the deposition of the aqueous suspension of NPs on top of the preformed monolayer is not typical; therefore, should be described in detail.

Approach 2

In the response to Reviewer’s remarks the authors provide a detailed description of their experimental procedure: “Mixed lipid/PHA monolayers were obtained by deposition of 15 µL of a mixture of 5 µL of the polymer dissolved in acetone, at different concentrations, with 995 µL of the chloroform/methanol lipid solution.” In my opinion all of this should be explicitly stated in the experimental part of the manuscript. Following my remarks the authors added a fragment regarding the effect of acetone on the characteristics of the DPPC isotherm. However, without the thorough description of the experimental procedure, and especially without the information that only 5 μL of acetone were mixed with 995 μL of chloroform the information of the acetone effects have no sense.

Please divide this fragment of experimental onto two separate paragraphs and describe thoroughly the first and the second approaches giving all the information necessary to follow the experimental procedure.

Remark 2

The COVASP LB approach to the Langmuir-Blodgett deposition is very rarely applied in experiments with Langmuir monolayers. As it was discussed in the ref. 29 it has its advantages and disadvantages as compared with the classical deposition of the film at a constant surface pressure value. In the first submission of the article this aspect of the experimental procedure was completely incomprehensible. Now adding the new paragraph and referring the readers to ref. 29 containing the detail description of the COVASP LB procedure the authors corrected this fragment considerably.

Remark 3

I am still very skeptical regarding the data presented in Fig. 2. I understand that these experiments were performed just to prove that the applied PHA NPs can interact with multulamellar vesicles (MLV) and change their characteristics. This fragment is very laconic, there is no in-depth discussion of these results provided by the authors. Moreover the results are not compared with any similar research – no reference is cited for this fragment. In the first review I asked the authors about the multimodal distribution of the MLV, especially regarding the DPPC MLVs and PL (0.7DPPC/0.3POPG) MLVs. I would like to state again that for DPPC and its mixture with POPG there in no physical reason for the multimodal distribution of the hydrodynamic radi of the MLVs. I am not satisfied by the authors’ response to this point. I can agree that such distributions can be possible for the natural mixtures containing SP proteins but not for MLVs composed of pure synthetic phospholipids. Moreover, the data presented in Fig. 2A including the multimodal distributions of MLVs should be discussed and crosslinked with other papers presenting similar research. Otherwise in my opinion the whole fragment should be deleted.

Moreover, this fragment should be linked to the further Langmuir results by an additional paragraph of the discussion. Here the presence of the NPs is obvious, for the Langmuir monolayers their presence is not obvious, as I already explained in the first review. How these results correlate with these obtained for Langmuir monolayer. How the results obtained for the MLV correlate with the potential pharmacological applications of PHA NPs?

Author Response

Remark 1

As suggested by the referee, we have included additional extensive details at the methodological section, to explain how (in a first approach) we carried out the experiments depositing different amounts of NPs on preformed phospholipid monolayers and how (second alternative approach) we tested the potential direct effect of the transfer of bioplastic polymer molecules into the lipid films.

We have also clarified that we tested different phospholipid/NP ratios (with no pre-established idea), and we selected 3 illustrative proportions to include into the different figures.

Remark 2

The referee is right that the COVASP method is not generally applied. However, we have been using it extensively since we developed it in 2007 (as described in reference 29). In our opinion is not used more frequently just because is not well known, because it provides much higher versatility. We have taken advantage of the comment by the referee to explain a bit more in the manuscript the advantage of using COVASP films, so that perhaps other readers could feel like testing and using it.

Remark 3

Following the suggestion by the reviewer, we have extended the description and interpretation of the DLS experiments, also comparing our results with those published by others on similar kinds of materials.

We have ben working extensively, over many years, with lipid suspensions and monolayers, using DLS, DSC and epifluorescence microscopy as fundamental techniques to characterize their behavior, and found that they complement very well. A key piece of information that connects in our opinion the information provided by DLS and Langmuir films is the existence of ordered (liquid-condensed in monolayers, gel in bilayers) phase segregation. Segregated regions define heterogeneity in both membrane and monolayer structures and in their mechanical properties (connecting with compression-driven collapse in monolayers, or with budding and vesiculation in liposomes), but also imply, in our opinion, differences with respect to the impact of the interaction with nanostructured bioplastic. We have now included a new paragraph in the discussion to integrate the different results in bilayers and monolayers with respect to the impact of the surfactant lipids/NPs interactions.

We are really grateful to the reviewer because with his/her input and the additional information and discussion incorporated, the paper has gained considerably in clarity and relevance.

Round 3

Reviewer 2 Report

The authors have followed the Reviewer's suggestions and corrected their manuscript considerably. In my opinion the manuscript is publishable in the present form.